# RGS7/Gβ5/R7BP complex regulates synaptic plasticity and memory by modulating hippocampal GABA_BR-GIRK signaling

Olga Ostrovskaya[1], Keqiang Xie[1], Ikuo Masuho[1], Ana Fajardo-Serrano[2], Rafael Lujan[2], Kevin Wickman[3], Kirill A Martemyanov[1]*

[1]Department of Neuroscience, The Scripps Research Institute, Jupiter, United States; [2]Departamento de Ciencias Médicas, Facultad de Medicina, Universidad de Castilla-La Mancha, Albacete, Spain; [3]Department of Pharmacology, University of Minnesota, Minneapolis, United States

**Abstract** In the hippocampus, the inhibitory neurotransmitter GABA shapes the activity of the output pyramidal neurons and plays important role in cognition. Most of its inhibitory effects are mediated by signaling from GABAB receptor to the G protein-gated Inwardly-rectifying K+ (GIRK) channels. Here, we show that RGS7, in cooperation with its binding partner R7BP, regulates GABA_BR-GIRK signaling in hippocampal pyramidal neurons. Deletion of RGS7 in mice dramatically sensitizes GIRK responses to GABA_B receptor stimulation and markedly slows channel deactivation kinetics. Enhanced activity of this signaling pathway leads to decreased neuronal excitability and selective disruption of inhibitory forms of synaptic plasticity. As a result, mice lacking RGS7 exhibit deficits in learning and memory. We further report that RGS7 is selectively modulated by its membrane anchoring subunit R7BP, which sets the dynamic range of GIRK responses. Together, these results demonstrate a novel role of RGS7 in hippocampal synaptic plasticity and memory formation.

*For correspondence: kirill@scripps.edu

**Reviewing editor**: Richard Aldrich, The University of Texas at Austin, United States

## Introduction

Signaling through G protein-coupled receptors for the inhibitory neurotransmitter GABA (GABA_BR) has been recognized to play key roles in mood, nociception, memory, reward, and movement (*Bowery, 2006*; *Padgett and Slesinger, 2010*). In the hippocampus, activation of postsynaptic GABA_BR on pyramidal neurons produces slow inhibitory postsynaptic currents (sIPSCs), which counteract the excitatory influence of ionotropic glutamate receptors to shape neuronal output (*Ulrich and Bettler, 2007*; *Luscher and Slesinger, 2010*). As a result, GABA_BR signaling profoundly affects hippocampal synaptic plasticity and has marked effects on memory formation (*Davies et al., 1991*; *Wagner and Alger, 1995*; *Schuler et al., 2001*).

A large share of the postsynaptic inhibitory effect of GABA_BR stimulation in the hippocampus is mediated by activation G protein-gated inwardly-rectifying K+ (GIRK/Kir3) channels, which inhibit neuronal excitability via hyperpolarizing K+ efflux (*Luscher and Slesinger, 2010*). In the hippocampus, GIRK channels are predominantly formed by GIRK1 and GIRK2 subunits, which co-localize and may interact directly with GABA_BR protomers (*Koyrakh et al., 2005*; *Fajardo-Serrano et al., 2013*). Activation of GABA_BR releases G protein βγ subunits, which bind to GIRK channels and increase channel gating (*Padgett and Slesinger, 2010*). Blockade of GABA_BR or GIRK channels by either pharmacological manipulations or genetic knockout ablates the slow IPSC, and blunts a form of hippocampal synaptic plasticity known as depotentiation (*Luscher et al., 1997*; *Chung et al., 2009*). Conversely, enhanced GABA_BR-GIRK signaling seen in a mouse model of Down syndrome disrupts

**eLife digest** Neurons communicate with one another at junctions called synapses. The arrival of an electrical signal known as an action potential at the first cell causes molecules known as neurotransmitters to be released into the synapse. These molecules diffuse across the gap between the neurons and bind to receptors on the receiving cell. Some neurotransmitters, such as glutamate, activate cells when they bind to receptors, thus making it easier for the second neuron to 'fire' (i.e., to generate an action potential). By contrast, other neurotransmitters, such as GABA, usually make it harder for the second neuron to fire.

Many of the effects of GABA involve a type of receptor called GABA$_B$. When GABA binds to one of these receptors, a molecule called a G-protein is recruited to the receptor. This activates the G-protein, triggering a cascade of events inside the cell that lead ultimately to the opening of potassium ion channels, which as known as GIRKs, in the cell membrane. Positively charged potassium ions then leave the cell through these channels, and this makes it more difficult for the cell to fire.

Now, Ostrovskaya et al. have revealed that a complex of three proteins regulates the interaction between GABA$_B$ receptors and GIRK channels. In neurons that lack either of these proteins, the receptors have less influence on GIRKs than in normal cells. Moreover, mice that lack one of the proteins (called RGS7) perform less well in various learning and memory tests: for example, they take longer than normal animals to learn the location of an escape platform in a water maze, or to retain a memory of a fearful event.

By identifying the proteins that regulate the interaction between GABA$_B$ receptors and GIRKs, Ostrovskaya et al. have helped to unravel a key signaling cascade relevant to cognition. Given that GIRK channels have recently been implicated in Down's syndrome, these insights may also increase understanding of cognitive impairments in neuropsychiatric disorders.

both excitatory and inhibitory synaptic plasticity, and is linked to cognitive impairment (*Kleschevnikov et al., 2004*; *Cramer et al., 2010*; *Cooper et al., 2012*).

GABA$_B$R-GIRK signaling is negatively modulated by the Regulators of G protein Signaling (RGS) proteins, which accelerate G protein inactivation (*Hollinger and Hepler, 2002*; *Padgett and Slesinger, 2010*). Among more than 30 RGS genes found in mammalian genomes, the R7 family of RGS proteins (R7 RGS) stands out for its prominent roles in a range of fundamental neuronal processes, from vision to motor control to reward-related behavior (*Anderson et al., 2009*). The four members of this group (RGS6, RGS7, RGS9 and RGS11) form heterotrimers with two subunits (Gβ5 and R7BP), and these interactions regulate the localization and/or expression of the complexes (*Anderson et al., 2009*; *Jayaraman et al., 2009*).

Previous studies have shown that Gβ5 serves as a central scaffold that bridges the catalytic (RGS) and targeting (R7BP) subunits and ensures the stability of R7 RGS protein (*Cheever et al., 2008*; *Sandiford et al., 2010*; *Masuho et al., 2011*). Elimination of Gβ5 also resulted in dramatic slowing of GIRK channel deactivation kinetics, prolongation of synaptically-evoked slow IPSCs in hippocampal pyramidal neurons, and increased behavioral sensitivity to GABA$_B$R stimulation (*Xie et al., 2010*). However, the identity of the RGS isoform that modulates GABA$_B$R-GIRK signaling in hippocampus, as well as the relative impact of the R7BP subunit, are unknown. Furthermore, the relevance of RGS-dependent modulation of GIRK-dependent signaling to hippocampal circuit function, plasticity, and behavior remain unclear.

In this study, we examined the importance of RGS and R7BP subunits to hippocampal physiology and hippocampal-dependent behavior. We report that ablation of *Rgs7* results in alterations of GABA$_B$R-GIRK signaling, disrupts synaptic plasticity in the hippocampus, and impairs contextual learning and memory. Moreover, the function of the RGS7/Gβ5 complex is fine-tuned by R7BP, which sets the sensitivity range of GABA$_B$R-GIRK signaling.

## Results

### RGS7 and R7BP modulate GABA$_B$R-GIRK signaling in cultured hippocampal neurons

We began by characterizing the expression of RGS complex subunits in the mouse hippocampus. We detected robust expression of RGS6, RGS7, R7BP and Gβ5 by western blotting (*Figure 1A*). To

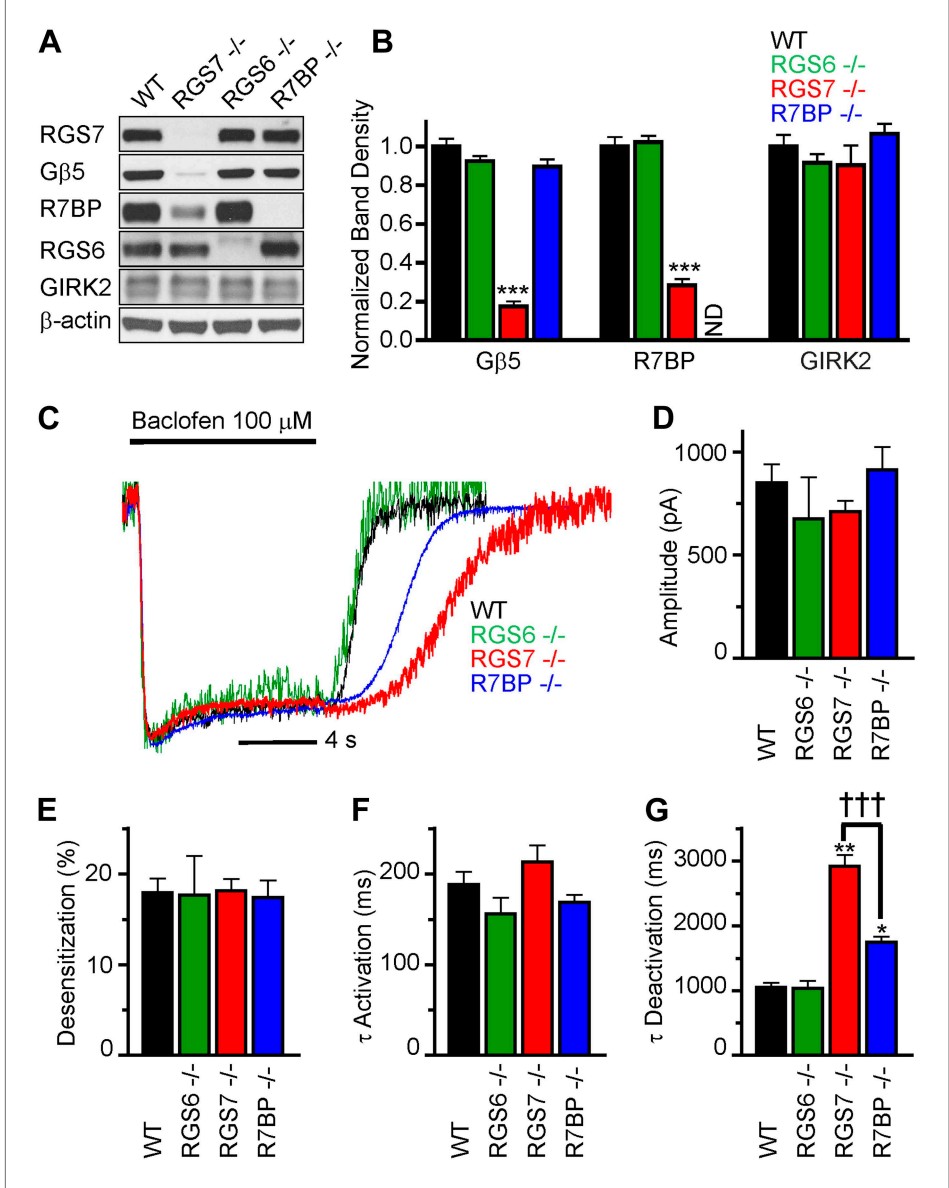

**Figure 1**. RGS7 and R7BP modulate GABA$_B$R-GIRK signaling in cultured hippocampal neurons. (**A**) Western blot analysis of protein expression in hippocampi extracted from wild-type (WT) mice, or mice lacking RGS7 (*Rgs7$^{-/-}$*), RGS6 (*Rgs6$^{-/-}$*) or R7BP (*R7bp$^{-/-}$*). (**B**) Quantification of Western blotting data, with protein levels arrayed as a function of genotype, ***p<0.01, One-Way ANOVA n = 3 mice. ND-undetectable. (**C**) Representative normalized traces of GIRK currents evoked by a saturating concentration of the GABA$_B$R agonist baclofen (100 µM). (**D–E**) *Rgs6, Rgs7, and R7bp* knockouts do not affect GIRK current amplitudes (**D**), desensitization (**E**) and current onset kinetics (**F**) evoked by 100 µM Baclofen. p>0.05, One-Way ANOVA, n = 8–29 cells for each genotype. (**G**) Current deactivation rate following removal of baclofen was slower in neurons from *Rgs7$^{-/-}$* and *R7bp$^{-/-}$* mice as compared to WT controls, *p<0.05 and **p<0.01 vs WT, †††p<0.001 *Rgs7$^{-/-}$* vs *R7bp$^{-/-}$*. One-Way ANOVA, Bonferroni's post hoc test, n = 8–29 cells.

understand the contribution of individual subunits to GABA$_B$R-GIRK signaling, we studied the effects of selective knockout of *Rgs6*, *Rgs7*, and *R7bp* in mice. Elimination of RGS7 dramatically reduced levels of Gβ5 and R7BP in the hippocampus (*Figure 1A,B*). In contrast, elimination of RGS6 had no significant effect on the expression of R7BP or Gβ5. Similarly, loss of one RGS protein did not affect the expression of the other, or the GIRK channel subunit GIRK2. Given the interdependence of subunit expression in R7 RGS complexes (*Chen et al., 2003*; *Anderson et al., 2007*; *Grabowska et al., 2008*), these results suggested that RGS7 was likely the dominant catalytic subunit in the hippocampus.

We next compared GABA$_B$R-GIRK responses in hippocampal pyramidal neurons from *Rgs6*, *Rgs7*, and *R7bp* knockout ($^{-/-}$) and wild-type (WT) mice (*Figure 1C–F*). Application of a saturating concentration (100 μM) of the GABA$_B$R agonist baclofen elicited currents with similar maximal amplitudes in neurons of all genotypes (*Figure 1D*). We also measured the activation and deactivation kinetics of the baclofen-induced currents. At this saturating concentration of baclofen, we observed no significant differences between genotypes in the activation phase of the response (*Figure 1C,F*). Desensitization of current during the timeframe of agonist application was negligible and similar across genotypes, suggesting that the response at the steady-state is not compounded by the GIRK channel inactivation (*Figure 1C,E*). While no change in the current deactivation kinetics was observed in *Rgs6*$^{-/-}$ neurons, elimination of RGS7 markedly slowed response deactivation (*Figure 1C,G*). The rate of GIRK current deactivation in *R7bp*$^{-/-}$ neurons was also slower than in wild-type neurons, but the effect was substantially smaller than seen in neurons from *Rgs7*$^{-/-}$ mice (*Figure 1C,G*). Importantly, there were no detectable changes in any of the measured response parameters in *Rgs6*$^{-/-}$ neurons as compared to wild-type, arguing that GABA$_B$R-GIRK signaling in hippocampal neurons is modulated by RGS7 and R7BP, but not RGS6.

## RGS7 and R7BP differentially affect the sensitivity of GABA$_B$R-GIRK coupling

Given the observed changes in the kinetics of GIRK channel modulation by GABA$_B$R, we next sought to determine the impact of ablating *Rgs7* and *R7bp* on the sensitivity of the GIRK response to GABA$_B$R activation. Increasing baclofen concentrations caused a progressive enhancement in GIRK-mediated currents in all genotypes (*Figure 2A*). However, pronounced differences in the concentration-response relationship were evident (*Figure 2B*). First, in both *Rgs7*$^{-/-}$ and *R7bp*$^{-/-}$ neurons, curves were shifted to the left relative to wild-type neurons, indicating that ablation of RGS7 or R7BP increased GABA$_B$R-GIRK coupling sensitivity. Second, while RGS7 ablation resulted in a largely parallel leftward shift of the curve, its shape in *R7bp*$^{-/-}$ neurons was markedly steeper. Indeed, response amplitudes resembled those seen in wild-type neurons at lower agonist concentrations, but at higher concentrations, the responses more resembled those seen in *Rgs7*$^{-/-}$ neurons.

We next tested whether GABA$_B$R-GIRK current kinetics exhibited a similar dependence on agonist concentration. Comparing activation rates across different concentrations of baclofen revealed that onset kinetics were affected by RGS7 ablation only at low baclofen concentrations, whereas no effect of R7BP elimination was seen at any level of GABA$_B$R stimulation (*Figure 2C*). Since GIRK activation kinetics could be influenced by the G protein deactivation cycle (*Doupnik et al., 1997*; *Lambert et al., 2010*), we did not put significant emphasis on the analysis of the differences in the raising phases of the response. The impact of RGS7 ablation on deactivation rates was consistent at different agonist concentrations (*Figure 2D*). In contrast, R7BP ablation significantly affected deactivation kinetics only at higher agonist concentrations, consistent with the effects on response sensitivity (*Figure 2D*).

Analysis of the responses also revealed that elimination of RGS7 and R7BP resulted in a significant delay between agonist removal and the beginning of current deactivation (*Figure 2E*). This lag time showed an exponential dependence on agonist concentration that continued to develop past the saturation point for the maximal GIRK response (~10 μM baclofen) in all genotypes (*Figure 2F*). While there was no difference in lag time between genotypes at non-saturating baclofen concentrations (~EC$_{60}$), it became pronounced as the current response reached saturation (~EC$_{90}$) (*Figure 2G*). Combined with the observations that GABA$_B$R continues to increase the amount of activated G proteins past the saturation point of the GIRK channel response (*Hensler et al., 2012*), these data suggest that the lag time reflects the clearance of free βγ subunit produced above the stoichiometric level relative to the GIRK channel.

To provide an independent evidence that increase in the lag time reflects changes in stoichiometry of Gβγ subunits relative to their effector molecule, we utilized a bioluminescence resonance energy transfer (BRET)-based approach that monitors interactions of Gβγ with a reporter derived from an effector (GRK3) upon reconstitution in transfected cells (*Figure 3A*). In this assay, we induced production of free Gβγ subunits via GABA$_B$R activation and then measured delay time between antagonizing GABA$_B$R and the onset of signal decay, while changing the Gβγ to GRK3 effector ratio (*Figure 3B*). In agreement with the electrophysiological recordings of GIRK channel activity, BRET experiments showed that the increase in Gβγ stoichiometry over an effector results in prolongation of the response deactivation lag (*Figure 3C,D*).

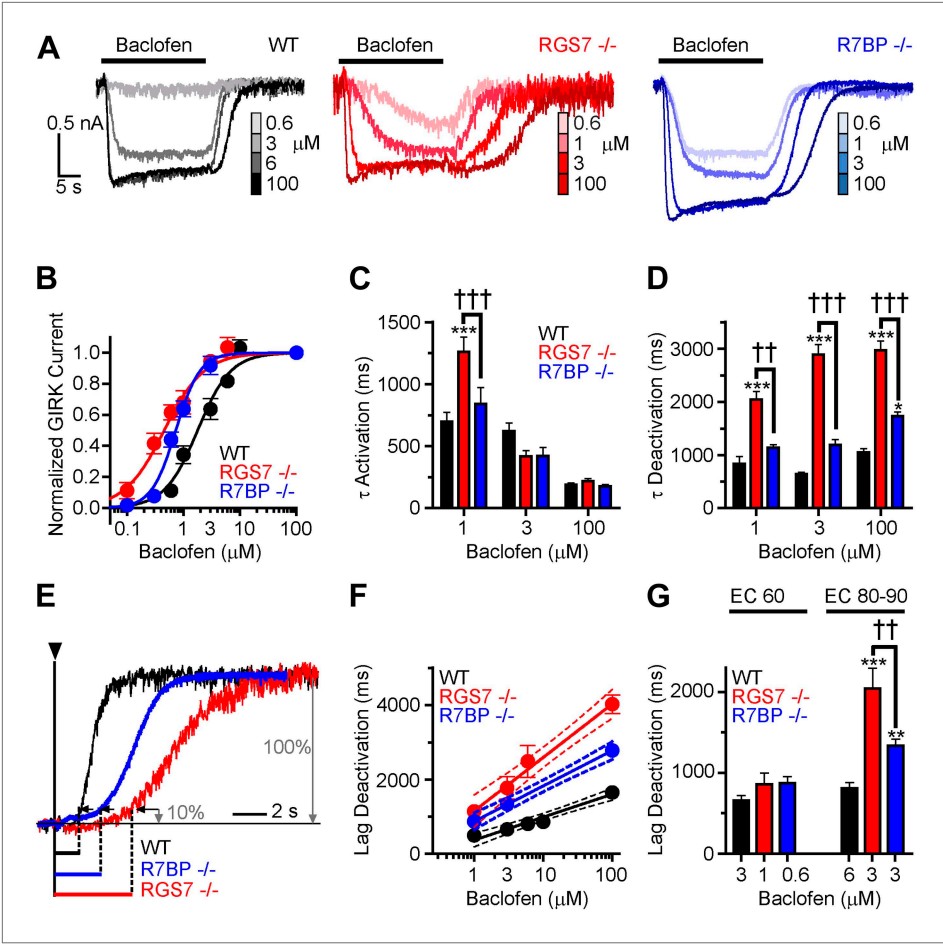

**Figure 2**. Timing and sensitivity of GABA$_B$R-GIRK signaling is differentially controlled by RGS7 and R7BP. (**A**) Representative traces of GIRK currents evoked by increasing concentrations of baclofen. (**B**) Dose-response curves fit by Hill equation. EC$_{50}$ values are 1.48–2.14 µM for WT, 0.35–0.58 µM for *Rgs7$^{-/-}$*, and 0.65–0.81 µM for *R7bp$^{-/-}$* (95% CI). Hill coefficients were 1.48 ± 0.17 in WT, 1.23 ± 0.21 in *Rgs7$^{-/-}$*, and 2.04 ± 0.28 in *R7bp$^{-/-}$*. p<0.0001 for difference in EC$_{50}$ or each curve; p=0.04 for difference in Hill coefficients, Extra sum-of-squares F test, n = 10–22 cells. (**C**) *Rgs7$^{-/-}$* cultured hippocampal neurons show slower current activation at lower concentrations of baclofen. ***p<0.001 vs WT, †††p<0.001 for *Rgs7$^{-/-}$* vs *R7bp$^{-/-}$*, Two-Way ANOVA, Bonferroni's post hoc test, n = 10–22 cells. (**D**) Dependence of GIRK current deactivation kinetics on agonist concentration. *p<0.05 and ***p<0.001 vs WT; ††p<0.01, †††p<0.001 for *Rgs7$^{-/-}$* vs *R7bp$^{-/-}$*, Two-Way ANOVA, Bonferroni's post hoc test, n = 10–22 cells. (**E**–**G**) Differences in the lag times before the onset of response deactivation (Lag Deactivation). (**E**) Representative traces of currents evoked by 100 µM baclofen show Lag time measurement as time between the onset of agonist removal and the point at which 10% of the current deactivated: 1560, 4900 and 3000 ms for WT, *Rgs7$^{-/-}$*, *R7bp$^{-/-}$*, respectively. (**F**) Semi-log plot of lag time dependence on concentration. The data were fitted with a linear regression, R$^2$ = 0.66, 0.67, and 0.84; and slopes 619 ± 69, 1440 ± 151, and 952 ± 84 for WT, *Rgs7$^{-/-}$*, *R7bp$^{-/-}$*, correspondingly. p<0.0001 for differences in slopes, two way ANOVA, n = 10–22 cells. (**G**) Comparison of lag times at baclofen concentrations that generated equivalent responses. Lag times were significantly different in genotypes at saturating (EC$_{90}$) but not submaximal (EC$_{60}$) concentrations, *p<0.05 and ***p<0.001 vs WT, ††p<0.01 *Rgs7$^{-/-}$* vs *R7bp$^{-/-}$*, Two-Way ANOVA, Bonferroni's post hoc test, n = 10–22 cells.

## Loss of R7BP reduces targeting of RGS7 to the plasma membrane

In transfected cells, R7BP is essential for the membrane localization of RGS7 (*Drenan et al., 2005*; *Narayanan et al., 2007*). Furthermore, we previously reported that knockout of *R7bp* resulted in a reduction of the total membrane-bound RGS7 protein in hippocampal tissue (*Panicker et al., 2010*). To analyze the effect of R7BP on RGS7 localization in hippocampal pyramidal neurons, we performed high-resolution immunoelectron microscopy. Consistent with the earlier findings, immunoparticles for

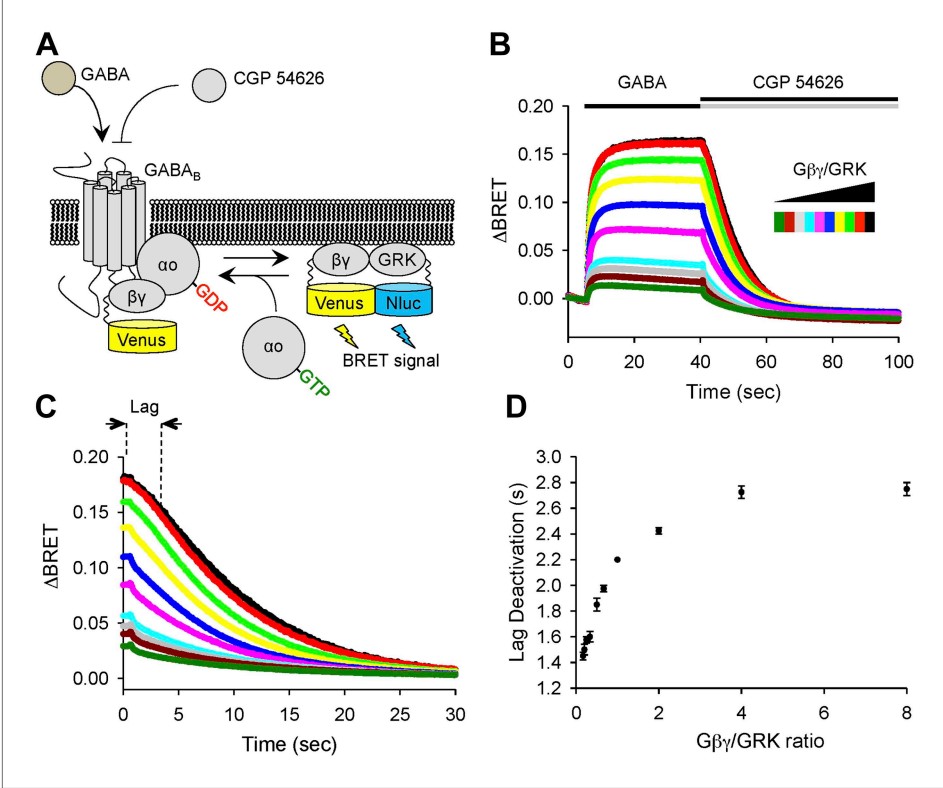

**Figure 3**. Changing the ratio of Gβγ to an effector affects response deactivation lag in a reconstituted system. (**A**) Schematic representation of the assay principle. The assay measures interaction of YFP-tagged Gβγ with its effector reporter GRK, tagged with N-luc, producing BRET signal. Gβγ subunits are released upon GABA$_B$R stimulation with GABA increasing BRET signal. Inactivation of GABA$_B$R with an antagonist CGP 54626 results in dissociation of Gβγ from GRK and re-association with Gαo to form inactive heterotrimer. (**B**) Time course of changes in BRET signal upon stimulation of cells with GABA and subsequent deactivation by CGP 54626. Cells were trans-fected with varying amounts of constructs encoding Gβγ and GRK reporter (from black to green). (**C**) Deactivation phase of the response showing kinetics of signal decay. The lag deactivation time (dotted line for the trace in black) is defined as the time that it takes to quench the BRET signal by 10% from its steady state value in the presence of an agonist. (**D**) Quantification of a lag deactivation time as a function of Gβγ/GRK ratio. Error bars are SEM values, n = 4 per condition.

RGS7 were abundant on the extrasynaptic plasma membrane of dendritic spines and dendritic shafts of pyramidal cells, as well as at intracellular sites (*Fajardo-Serrano et al., 2013*). In the hippocampus of *R7bp$^{-/-}$* animals, immunoparticles for RGS7 were distributed similarly to the wild-type, but they were more frequently observed just beneath the plasma membrane and more broadly distributed in somata (*Figure 4*). Indeed, quantitative analysis indicated that in *R7bp$^{-/-}$* mice, RGS7 was less frequently detected in the plasma membrane and tended to accumulate within 100 nm of the plasma membrane. We also detected a significant increase in RGS7 labelling in the rough endoplasmic retic-ulum (rER) in the soma of *R7bp$^{-/-}$* pyramidal neurons (2489 vs 3371 immunoparticles in wild-type and *R7bp$^{-/-}$* neurons, respectively). These findings indicate that knockout of R7BP caused a modest but significant redistribution of RGS7 away from the plasma membrane.

## Loss of RGS7 decreases the excitability of hippocampal pyramidal neurons

The slower kinetics and increased sensitivity of GIRK channels to GABA$_B$R activation suggested that there was a net up-regulation of GIRK-dependent inhibition in hippocampal neurons lacking RGS7. Since GIRK channels significantly contribute to setting neuronal excitability (*Chen and Johnston, 2005*), we next determined how deletion of RGS7 influences the excitability of CA1 pyramidal neu-rons. To characterize intrinsic electrophysiological properties of different genotypes, the responses of

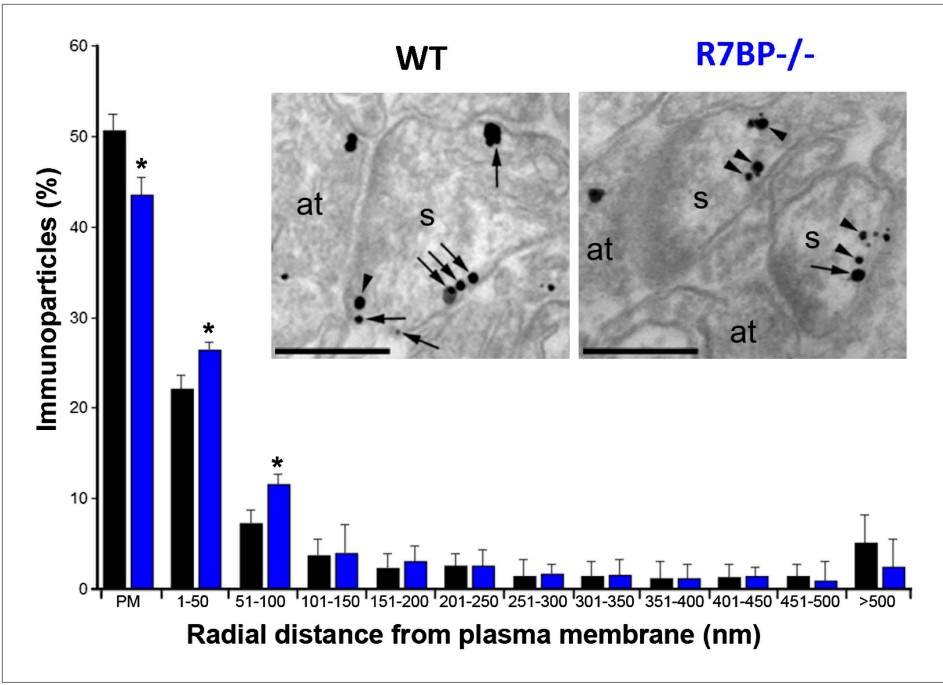

**Figure 4**. Change in subcellular localization of RGS7 in the hippocampus of the *R7bp*<sup>−/−</sup> mice. Electron micrographs of the *stratum radiatum* of the hippocampal CA1 region showing immunoparticles for RGS7, as detected using a pre-embedding immunogold method. Dendritic spines (s) and axon terminals (at) are marked. Arrows indicate locations of immunoparticles at the plasma membrane, while arrowheads identify RGS7 immunoparticles found just below the membrane. Quantitative analysis showed that RGS7 is less frequently detected in the plasma membrane, and accumulates within the first 100 nm from the plasma membrane, samples from *R7bp*<sup>−/−</sup> mice, *p<0.05, One-way ANOVA followed by the Bonferroni's post hoc test, n = 3 mice. Scale bar: 0.2 μm.

CA1 neurons to somatic current injections ranging from −150 pA to +300 pA were measured (*Figure 5A*). While membrane resistance was similar in both genotypes (Rin = 120 ± 10 MΩ vs 105 ± 11 MΩ, p=0.31, in wild-type and *Rgs7*<sup>−/−</sup> cells, respectively), resting membrane potential (RMP) was significantly hyperpolarized in *Rgs7*<sup>−/−</sup> neurons (−64.7 ± 0.5 mV vs −68.5 ± 0.7 mV in wild-type and *Rgs7*<sup>−/−</sup>, respectively; *Figure 5B*). In addition, the current required to elicit action potentials (APs) was significantly higher in *RGS7*<sup>−/−</sup> neurons (I = 104.3 ± 8.8 pA vs 146.4 ± 17.0, in WT and *Rgs7*<sup>−/−</sup> cells, respectively; *Figure 5C*). Finally, *Rgs7*<sup>−/−</sup> neurons fired significantly fewer action potentials in response to depolarizing current as compared to wild-type controls (*Figure 5D*). Collectively, these observations argued that hippocampal pyramidal neurons from *Rgs7*<sup>−/−</sup> mice were less excitable than wild-type counterparts.

We further examined whether decreased intrinsic excitability of *Rgs7*<sup>−/−</sup> neurons is caused by dysregulated net excitatory transmission. For this purpose, we measured spontaneous EPSCs (sEPSC) in CA1 pyramidal neurons (*Figure 5E–G*). There were no significant differences in either amplitudes or frequencies of sEPSC events between *Rgs7*<sup>−/−</sup> and wild-type neurons (*Figure 5F–G*). These observations indicate that excitatory input into CA1 pyramidal neurons is unchanged by elimination of RGS7 and suggest that the observed decrease in neuronal excitability is caused by changes in the intrinsic membrane properties of CA1 neurons, likely stemming from enhanced activity of the postsynaptic GIRK channel.

## RGS7 ablation disrupts hippocampal synaptic plasticity

Previous studies have implicated GIRK channels in several forms of hippocampal synaptic plasticity, including long-term potentiation (LTP) (*Cramer et al., 2010*), long-term depression (LTD) (*Cooper et al., 2012*) and depotentiation (DP) (*Chung et al., 2009*; *Cooper et al., 2012*). Because GABA<sub>B</sub> receptors play key roles in these processes and since GABA<sub>B</sub>-GIRK signaling is severely dysregulated in *Rgs7*<sup>−/−</sup> neurons, we next examined the impact of RGS7 ablation on hippocampal synaptic plasticity.

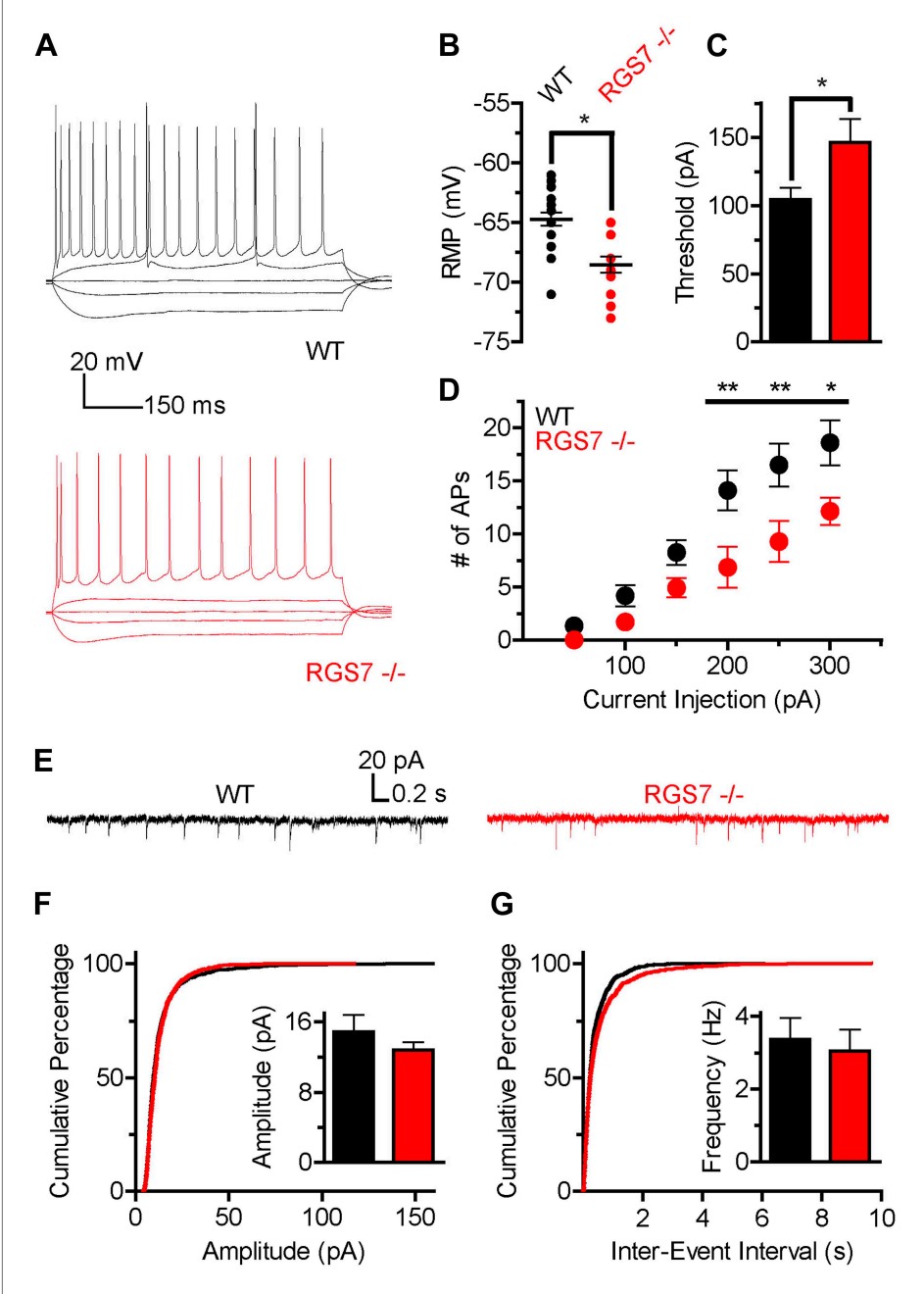

**Figure 5**. Altered intrinsic excitability and excitatory transmission in CA1 hippocampal neurons from *Rgs7−/−* mice.
(**A**) Representative traces of responses elicited by current injections of −150, −50, 0, +50, and +300 pA in WT and
*Rgs7−/−* CA1 neurons. (**B**) Hyperpolarized resting membrane potential (RMP) in *Rgs7−/−* neurons, ***p=0.0001, *t* test,
n = 14–23 cells. (**C**) Current required to evoke an action potential (firing threshold) is higher in *Rgs7−/−* neurons,
*p=0.02, *t* test, n = 14–23 cells. (**D**) Lower intrinsic excitability in *Rgs7−/−* neurons. **p<0.01 and *p<0.05, two-way
ANOVA with Bonferroni's posttest, n = 14–23 cells. (**E**) Representative traces of slow excitatory synaptic currents
(sEPSCs) in WT and *Rgs7−/−* cells. (**F**) Cumulative distribution and mean values for sEPSCs amplitudes (14.9 ± 1.9 vs
12.8 ± 0.9 pA in WT and *Rgs7−/−* correspondingly, p=0.3, unpaired *t* test; n = 11 cells and 1100 events for each
genotype). (**G**) Cumulative distribution and mean values for sEPSCs and frequencies (3.4 ± 0.6 vs 3.1 ± 0.6 Hz in WT
and *Rgs7−/−* correspondingly, p=0.7, unpaired *t* test, n = 11 cells and 1100 events for each genotype).

A high-frequency stimulation protocol of 2 tetanized stimuli (TS, 100 Hz for 1s each) produced robust LTP in both genotypes (*Figure 6A,D*). The extent of the potentiation in *Rgs7*⁻/⁻ slices was not significantly different from that measured in wild-type slices, arguing that elimination of RGS7 has no effect on LTP. A low-frequency stimulation (LFS, 2 Hz for 10 min, 1200 pulses) elicited LTD in both wild-type and *Rgs7*⁻/⁻ hippocampal slices (*Figure 6B,D*). However, the extent of inhibition was significantly smaller in *Rgs7*⁻/⁻ slices. Interestingly, the magnitude of the first *f*EPSP after the train of LFS stimulation was significantly reduced in knockout mice (51 ± 6% and only 72 ± 6% in wild-type and *Rgs7*⁻/⁻ slices correspondingly, *p=0.03, *t* test, n = 6), suggesting that the LTD impairment in *Rgs7*⁻/⁻ is

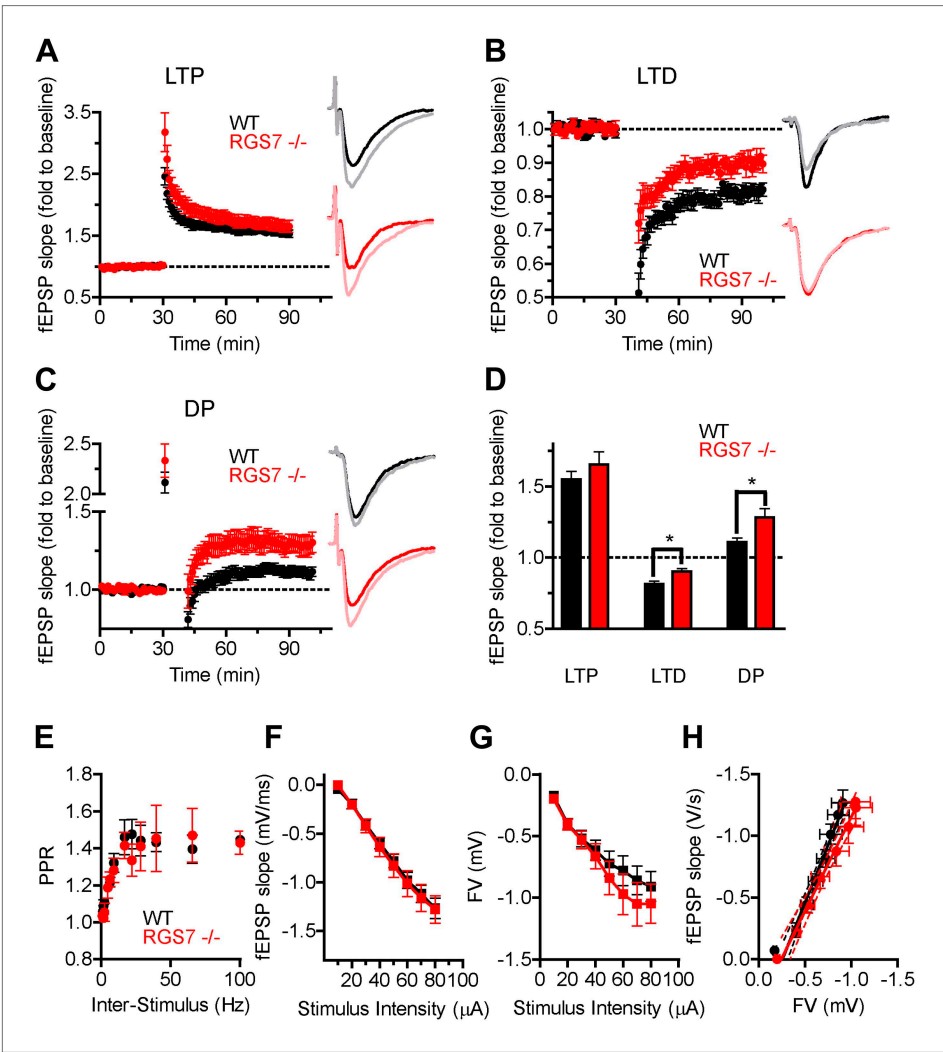

**Figure 6**. RGS7 ablation disrupts hippocampal synaptic plasticity. Field EPSP (*f*EPSP) slope change following induction of: (**A**) LTP in *Rgs7*⁻/⁻, 166 ± 9% vs wild-type, 155 ± 6% slices; p=0.33, *t* test, n = 6–10; (**B**) LTD in *Rgs7*⁻/⁻, 90 ± 2% vs wild-type, 81 ± 2%; *p=0.01, *t* test, n = 6–7; (**C**) depotentiation (DP) in *Rgs7*⁻/⁻, 111 ± 3% vs wild-type, 128 ± 6%; *p=0.013, *t* test, n = 6–11. Insets show representative *f*EPSP traces at baseline and 1 hr following induction protocol in WT (black and grey) and *Rgs7*⁻/⁻ (red and pink) slices. (**D**) Quantification of the EPSP slope change 55–60 min following induction of each form of plasticity after normalization to pre-induction baseline. (**E**) Paired Pulse Ratio (PPR) dependence on the inter stimulus interval for wild-type (WT) and *Rgs7*⁻/⁻. Significant inter-stimulus interval but not genotype effect was observed, p<0.0001 and p=0.5, correspondingly. Two-way ANOVA, n = 4 slices per genotype. (**F–H**) Basal synaptic transmission properties in *Rgs7*⁻/⁻ slices. Dependence of *f*EPSP slope (**F**) and FV amplitude (**G**) on stimulus intensity. (**H**) Linear regression plot of *f*EPSP slope dependence on FV amplitude. The data were fitted with a linear regression, $R^2$ = 0.67 and 0.53; and slopes 1.9 ± 0.1 and 1.5 ± 0.2 for WT and *Rgs7*⁻/⁻correspondingly. p=0.1 for differences in slopes, two way ANOVA, n = 12–13 cells.

due to a deficit in an induction mechanism(s). The depotentiation produced by consecutive application of TS and LFS was also significantly impaired in $Rgs7^{-/-}$ mice (**Figure 6C,D**).

We further investigated the impact of RGS7 ablation on paired-pulse facilitation (PPF), a form of short-term plasticity that results from the enhancement of presynaptic vesicle release in response to two closely-spaced stimuli (**Zucker, 1989**; **Dobrunz and Stevens, 1997**). We found that fEPSP facilitation was similar between genotypes across the examined range of 10–1000 ms interpulse intervals (**Figure 6E**), indicating that presynaptic mechanisms are not likely to be the cause of the observed alterations in synaptic plasticity. We next compared the dependence of fEPSP slopes (**Figure 6F**) and presynaptic fibre volley (FV) amplitudes (**Figure 6G**) on stimulus intensity. Analysis of the relationship between FV amplitudes and fEPSP slopes by linear fitting reveals their nearly perfect correspondence between wild-type and $Rgs7^{-/-}$ slices, indicating preservation of basal synaptic transmission properties (**Figure 6H**). Together, these results suggest that RGS7 elimination selectively impaired LTD and DP forms of synaptic plasticity, likely by a post-synaptic mechanism.

## RGS7 ablation disrupts hippocampal-dependent learning and memory

Changes in the intrinsic excitability of CA1 pyramidal neurons, together with deficits in hippocampal synaptic plasticity, suggested that RGS7 complexes play a role in spatial learning and memory. To test this possibility, we first studied the impact of RGS or R7BP ablation in contextual fear conditioning, a test that requires hippocampal processing for memory formation (**Maren, 2001**). We found significant deficits in context recognition 24-hr after training in $Rgs7^{-/-}$ mice, but not in $Rgs6^{-/-}$ or $R7bp^{-/-}$ mice (**Figure 7A–C**). Importantly, we observed no difference in baseline freezing behavior before associative training between the genotypes. Furthermore, there were no significant differences between wild-type mice and mutant mice in amygdala-dependent cue recognition.

To clarify the impairment of hippocampus dependent learning and memory in $Rgs7^{-/-}$ mice, we conducted other tests that rely on hippocampal function. In the Morris water maze test, $Rgs7^{-/-}$ mice showed delayed escape latencies and reduced success rates during acquisition trials (**Figure 7D,E**). In the probe trial, when the platform was removed 24 hr after training, it took $Rgs7^{-/-}$ mice significantly longer to reach the target area where the platform was located during training as compared to their wild-type littermates (**Figure 7F**). Mice lacking RGS7 also made fewer crossings over the target area during the probe trial. Importantly, despite a pronounced spatial impairment, $Rgs7^{-/-}$ mice performed normally in the visible platform version of water maze.

In the novel object recognition test, wild-type littermates spent significantly more time exploring novel objects over familiar objects (**Figure 7G**). However, $Rgs7^{-/-}$ mice spent approximately equal time exploring familiar and novel objects. Furthermore, there was a significant difference in time exploring novel objects between $Rgs7^{-/-}$ mice and their wild-type littermates. No genotype differences in object exploratory behavior were observed during the training day, when the mice were first introduced to identical objects. Taken together, these results indicate that elimination of RGS7 leads to a disruption of spatial learning and memory in mice.

## Discussion

### Molecular mechanism of GIRK channel regulation by the RGS7/Gβ5/ R7BP complex

The results of this study, together with prior investigations, establish the RGS7/Gβ5/R7BP complex as an essential regulator of $GABA_BR$-GIRK signaling in the hippocampus. In hippocampal neurons, RGS7 is closely co-localized with both $GABA_BR$ and GIRK2-containing channels (**Fajardo-Serrano et al., 2013**). Furthermore, in transfected cells, RGS7 can directly interact with GIRK channel subunits as demonstrated by both co-immunoprecipitation and bioluminescence energy transfer approaches (**Xie et al., 2010**; **Zhou et al., 2012**). The complex may further involve $GABA_BR$ as well as G protein subunits (**Kovoor and Lester, 2002**; **Fajardo-Serrano et al., 2013**), supporting the contention that components of the entire pathway are scaffolded into a larger macromolecular assembly.

The interaction between RGS7 with GIRK channels is mediated by the Gβ5 subunit, which mimics signal-transducing Gβγ subunits in binding to the channel (**Xie et al., 2010**; **Zhou et al., 2012**). Knockout of Gβ5 greatly slows $GABA_BR$-GIRK response deactivation rates and dramatically increases agonist sensitivity of the signaling pathway (**Xie et al., 2010**). In addition to mediating interactions with the channel, Gβ5 is essential for the expression of all four R7 RGS proteins (**Chen et al., 2003**).

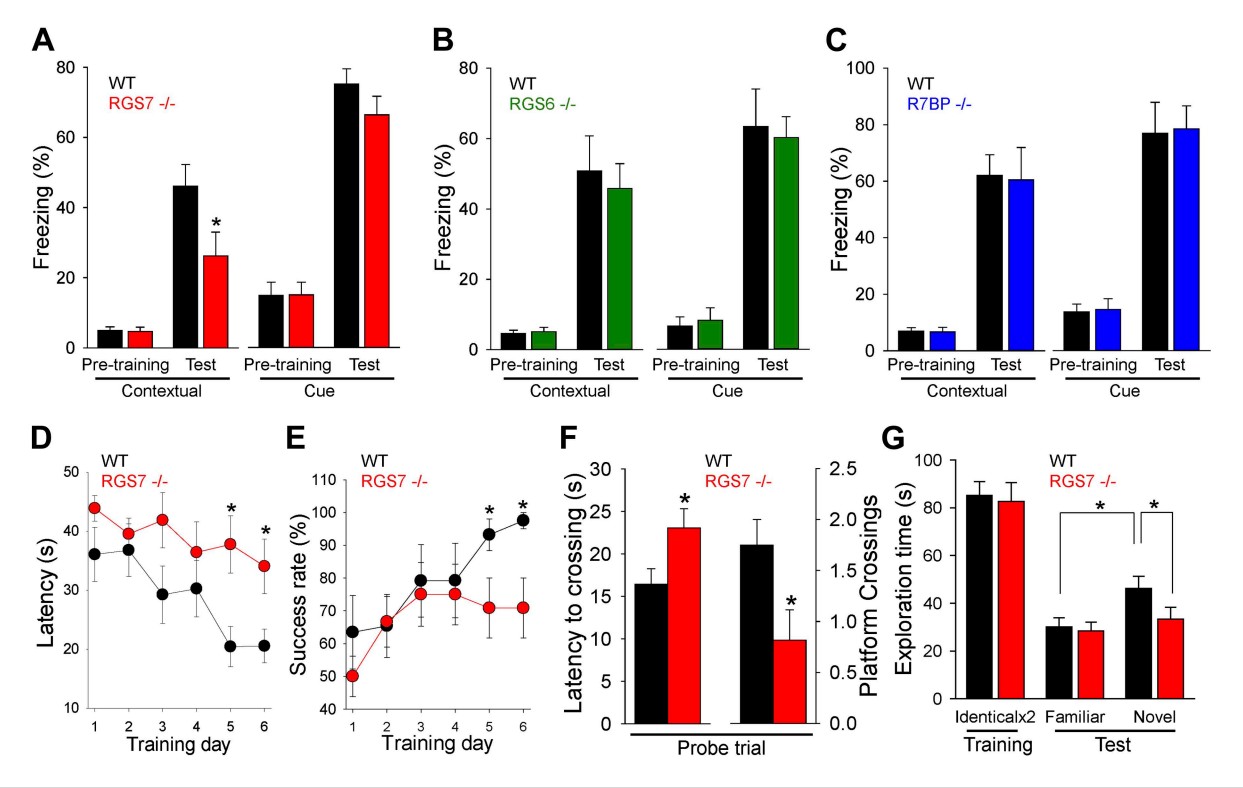

**Figure 7**. RGS ablation affects hippocampal-dependent learning and memory. (**A–C**) Evaluation of mouse behavior in fear conditioning paradigm. (**A**) *Rgs7*−/− mice show deficits in hippocampal-dependent contextual, but not cue, memory.*p<0.05, *t* test; n = 12 per genotype. (**B**) Normal contextual and cue memory of *Rgs6*−/− mice in the fear conditioning test as compared to wild-type (WT) littermates. n = 8 per genotype. (**C**) *R7bp*−/− mice showed the same contextual and cue memory in fear conditioning test as compared to WT littermates, n = 8 per genotype. (**D–F**) Evaluation of mouse behavior in Morris water maze. *Rgs7*−/− mice (n = 13) and their WT littermates (n = 12) were trained for 6 d with 4 trials/d with an inter-trial interval of approximately 15 min. Performance during the acquisition phase was monitored and plotted as average time (**D**) or success rate (**E**) to reach the hidden platform. Mice showed improvement with training. There was a significant effect of a genotype in both escape latencies (*p<0.05) and success rates (*p<0.05) using two-way ANOVA analysis. *Post hoc* comparison revealed a significant impairment of *Rgs7*−/− mice during the last two acquisition days (*p<0.05, Tukey's test). (**F**) Results of a probe trial given 24 hr after 6 d of training. The latency to the first crossing of the former location of the platform and the total number of crossing are shown, *p<0.05, *t* test. (**G**) Evaluation of mouse behavior in novel object recognition paradigm. *Rgs7*−/− mice showed significant impairment during the test trial with a novel object in comparison with wild-type littermates. *p<0.05, *t* test; n = 12–13.

With several R7 RGS proteins expressed in hippocampus, the identity of the exact isoform that participates in GIRK channel regulation in the region, through complex formation with Gβ5, remained unknown. The elimination of RGS7, reported in this study, largely phenocopies the loss of Gβ5, both in terms of kinetics and agonist sensitivity. This argues that in hippocampal pyramidal neurons, GIRK signaling is regulated by RGS7 and not by other R7 RGS proteins. Indeed, while RGS6 regulates GIRK signaling in cerebellar neurons (***Maity et al., 2012***) and sinoatrial pacemaking cells (***Posokhova et al., 2010***; ***Yang et al., 2010***), RGS6 ablation had no effect on any GIRK response parameters in hippocampal pyramidal neurons.

The function of R7 RGS proteins is regulated by the membrane anchor R7BP, which augments the ability of R7 RGS proteins to deactivate G protein signaling in reconstituted systems (***Drenan et al., 2006***; ***Masuho et al., 2013***). Consistent with these observations, we show that knockout of R7BP reduces plasma membrane localization of RGS7 and slows GABA$_B$R-GIRK current deactivation in hippocampal pyramidal neurons. Our results also agree well with a recent study reporting similar deceleration of GIRK deactivation kinetics in *R7bp*−/− hippocampal neurons (***Zhou et al., 2012***). However, the consequences of RGS7 and R7BP elimination are different in two important ways. First, relative to the loss of RGS7 or Gβ5, elimination of R7BP had a very modest effect on GABA$_B$R-GIRK deactivation kinetics, and was seen only at high GABA$_B$R agonist concentrations. This suggests that the RGS7/Gβ5

complex can regulate GABA$_B$R-GIRK signaling without R7BP. Secondly, loss of R7BP and RGS7 has different effects on GABA$_B$R-GIRK coupling sensitivity. While elimination of RGS7 (or Gβ5) results in a parallel leftward shift in the concentration-response relationship, the concentration-response measured in *R7bp$^{-/-}$* neurons shows markedly greater cooperativity. Thus, at low levels of GABA$_B$R stimulation, R7BP is dispensable and its elimination fails to affect response amplitude. As the response reaches saturation, R7BP becomes indispensable and its elimination has the same effect on response amplitude as loss of RGS7. These observations are inconsistent with the proposed role for R7BP as a critical factor in the assembly of RGS7-GIRK complexes (*Zhou et al., 2012*), which postulates the equivalence of R7BP effects across agonist concentrations and a similar impact of R7BP and RGS7 elimination on GIRK channel kinetics, and certainly not an increase in cooperativity of GIRK channel activation.

Based on our findings, we propose an alternative model, where R7BP sets GABA$_B$R-GIRK coupling efficiency. In this model, at low levels of GIRK activation, when released Gβγ subunits do not saturate the GIRK channel, deactivation kinetics are primarily mediated by RGS7/Gβ5 directly associated with the GIRK subunits. Under these conditions, there are virtually no genotype-dependent differences in the lag time of GIRK deactivation upon agonist removal. With higher agonist concentrations, as GABA$_B$R produces more Gβγ subunits, the amount of available free Gβγ exceeds that of activatable GIRK channels, evidenced by a delay that precedes GIRK current deactivation upon agonist removal. Under these conditions, R7BP elimination affects response sensitivity as much as elimination of RGS7. While response deactivation kinetics reflect Gβγ inactivation after it had a chance to interact with the channel, response sensitivity likely relates to the efficiency of the Gβγ reaching the GIRK channel. Considering that R7BP affects both of these parameters only when free Gβγ is produced in excess, we think that the RGS7/Gβ5 complex exists in two states: (1) anchored to the GIRK complex where it affects deactivation kinetics, and (2) anchored via R7BP to the plasma membrane outside of GIRK complex where it primarily determines response sensitivity.

In agreement with this model, we observe an increase in cooperativity of GIRK channel activation upon R7BP loss that suggests that RGS7/Gβ5 alone, but not in complex with R7BP, promotes more productive association of Gβγ with the channel. On an intuitive level, this may reflect larger proportion of Gβγ subunits that reach GIRK while being deactivated in its vicinity by RGS7/Gβ5. In contrast, complexes of RGS7/Gβ5 with R7BP may predominantly act elsewhere on the plasma membrane, thereby decreasing the number of Gβγ subunits capable of reaching the channel. By physically binding to the GIRK channel, RGS7/Gβ5 may further act to promote productive interactions of Gβγ (and/or Gα) with the channel, thus increasing cooperativity of its activation. This model is also consistent with the observed decrease in RGS7 on the plasma membrane but largely preserved localization at specific postsynaptic sites where it might be anchored via complex formation with GIRK channels. Thus, the main role of R7BP appears to be in tuning the sensitivity of the response by endowing RGS7/Gβ5 complexes with an ability to deactivate G proteins before they reach the GIRK channel.

## RGS7 is a new player in learning and memory and synaptic plasticity

RGS proteins are potent negative regulators of both the extent and duration of neurotransmitter signaling via G protein-coupled receptors (*Sjogren, 2011*; *Xie and Martemyanov, 2011*). As a result, knockout of individual RGS genes in mice is frequently associated with augmented GPCR signaling. This makes the analysis of changes in sensitivity of behavioral or cellular reactions to neurotransmitter actions associated with elimination of individual RGS proteins a powerful strategy that allows for establishing physiological receptor-RGS pairings. Using this strategy, two RGS proteins have been previously implicated in controlling GABA$_B$R signaling in vivo: RGS6 in cerebellar neurons (*Maity et al., 2012*) and RGS2 in the ventral tegmental area (*Labouebe et al., 2007*). Now, our findings establish that in hippocampal neurons, GABA$_B$R signaling is negatively regulated by RGS7. These observations complement earlier findings with RGS14, the only other RGS protein implicated in hippocampal synaptic plasticity and spatial learning (*Lee et al., 2010*). However, in contrast to RGS7 that acts in the CA1 region, RGS14 functions in CA2 synapses and its ablation augments LTP and improves spatial learning in mice.

The major forms of synaptic plasticity (LTP, LTD and depotentiation) in the hippocampus have been implicated in learning and memory (*Martin et al., 2000*). Recent models propose that all three forms cooperate to affect distinct aspects of spatial information storage (*Kemp and Manahan-Vaughan, 2007*). In relevance to this thinking, we found a selective disruption of LTD and depotentiation in *RGS7$^{-/-}$* mice, but normal LTP. There is a mounting evidence suggesting an active and selective role of

LTD in object recognition and creation of spatial representation of memory (*Kemp and Manahan-Vaughan, 2007*). For example, LTD but not LTP was found to be important for spatial memory consolidation and memory enhancement for novelty acquisition (*Ge et al., 2010*; *Dong et al., 2012*). Consistent with this, electrophysiological recordings in behaving animals show facilitation of LTD during novelty exploration (*Xu et al., 1997*, *1998*; *Manahan-Vaughan and Braunewell, 1999*; *Dong et al., 2012*). Disruption of LTD via genetic mutations often results in memory deficits, particularly affecting behavioral flexibility (*Nicholls et al., 2008*), while LTD enhancement can lead to improvement in spatial reversal learning (*Duffy et al., 2008*). In particular, our findings resemble the phenotype of SRF$^{-/-}$ mice, which exhibited a selective disruption of LTD in hippocampal CA1 neurons paralleled by an inability to learn hippocampus-dependent tasks (*Etkin et al., 2006*). While specific knowledge on how depotentiation contributes to memory is still lacking, it has been observed that exposure to novel environment and spatial exploration depotentiates previously-induced LTP (*Xu et al., 1998*). Together, our observations in mice lacking RGS7 reinforce the idea that normal inhibitory synaptic plasticity (LTD and depotentiation) is required for hippocampal-dependent learning and memory.

Although, it has never been systematically addressed in a single study, GIRK channel elimination was reported to enhance LTP (*Cramer et al., 2010*). Given that the *Girk2*$^{-/-}$ neurons are relatively depolarized, the enhanced LTP likely results from a general increase in excitability. Conversely, GIRK2 over-expression in transgenic models of Down syndrome leads to reduced LTP (*Kleschevnikov et al., 2004*; *Siarey et al., 2005*), but enhanced LTD (*Siarey et al., 2005*; *Cooper et al., 2012*). Interestingly, GIRK2 over-expression and ablation both resulted in decreased depotentiation (*Chung et al., 2009*; *Cooper et al., 2012*).

Similar to Down syndrome mouse models, *Rgs7*$^{-/-}$ mice also exhibited augmented GABA$_B$R-GIRK signaling. However, in the case of *Rgs7*$^{-/-}$ mice, this augmentation resulted in yet another distinct phenotype: selective deficits in LTD and depotentiation, with normal LTP. Several lines of evidence argue that these changes in synaptic plasticity are driven by a postsynaptic mechanism whereby RGS7 controls inhibitory GIRK-dependent signaling in the CA1 pyramidal neurons. First, CA1 pyramidal neurons lacking RGS7 are hyperpolarized and less excitable relative to wild-type neurons, a property that is significantly shaped by the GIRK channels. Second, basal excitatory transmission is unaltered in *Rgs7*$^{-/-}$ slices. Third, direct measurements of events that reflect excitatory presynaptic function (e.g., FV and PPR) reveal no changes caused by the elimination of RGS7. Given these observations, we propose that GIRK signaling sets the general excitability of postsynaptic pyramidal neurons, which in turn determines a sliding scale window for the induction of different forms of synaptic plasticity. By adjusting signaling strength in the hippocampal GABA$_B$R-GIRK pathway, the RGS7/Gβ5/R7BP complex influences the range of neuronal responses necessary for memory formation. While dysregulation of GABA$_B$R-GIRK signaling in hippocampus may be sufficient for explaining the effects of RGS7 on learning an memory, at this point we cannot rule out a contribution of other brain regions and/or signaling pathways in the process. In any event, we believe that the results of this study establish RGS7 complex as an important molecule for understanding and/or correcting the pathology of neuropsychiatric disorders associated with disruptions in synaptic plasticity and imbalances in inhibitory signaling.

## Materials and methods

### Animals

All studies were carried out in accordance with the National Institute of Health guidelines and were granted formal approval by the Institutional Animal Care and Use Committee of the Scripps Research Institute. The generation of *Rgs6*$^{-/-}$ (*Posokhova et al., 2010*), *Rgs7*$^{-/-}$ (*Cao et al., 2012*), *R7bp*$^{-/-}$ (*Anderson et al., 2007*) mice were described earlier. All animals used for comparing genotypes were littermates derived from heterozygous breeding pairs. Mice were housed in groups on a 12 hr light–dark cycle with food and water available *ad libitum*. Males and females (2–5 months) were used for all experiments.

### Antibodies and western blotting

Lysates were prepared by homogenizing hippocampal tissue from age-matched littermates by sonication in the lysis buffer (1 × PBS, 150 mm NaCl, 1% Triton X-100, protease inhibitors) followed

centrifugation at 14,000×$g$ for 10 min. The resulting extract was used for protein concentration determination by the BCA protein assay (Pierce, Rockford, IL). The lysates were adjusted to equalize total protein content by adding lysis buffer and 2x SDS sample buffer. Samples were boiled for 5 min, resolved on SDS-PAGE gels, transferred onto PVDF membrane and subjected to western blot analysis using HRP conjugated secondary antibodies and ECL West Pico (Pierce) detection system. Signals were captured on film and scanned by densitometer, and band intensities were determined using NIH ImageJ software. Rabbit anti-R7BP (TRS) and rabbit Gβ5 (ADTG) were generous gifts from Dr William Simonds (NIDDK/NIH). Anti-RGS6 was generated and used previously (*Posokhova et al., 2010*). Chicken IgY anti-RGS7 antibody was from Thermo Scientific (Waltham, MA), anti-GIRK2 was purchased from Alomone labs (Jerusalem, Israel) and anti-β-actin was from Sigma (St. Lois, MO).

## Hippocampal cultures

Primary cultures of hippocampal neurons were prepared using a modified version of a published protocol (*Xie et al., 2010*). Briefly, hippocampi were extracted from neonatal (P1-3) pups and placed into an ice-cold HBSS/FBS solution: Hank's Balanced Salt Solution (Sigma; St. Louis, MO), 4.2 mM NaHCO$_3$, 1 mM HEPES, and 20% FBS. The tissue was washed twice with HBSS/FBS, and then three times with HBSS alone. Hippocampi were digested at room temperature for 5 min with 10 mg/ml Trypsin Type XI (Sigma; St. Louis, MO) in a solution that contained (in mM): 137 NaCl, 5 KCl, 7 Na$_2$HPO$_4$, and 25 HEPES (pH 7.2). The tissue was washed twice with HBSS/FBS and three times with HBSS alone, and then hippocampi were mechanically-dissociated in HBSS (supplemented with 12 mM MgSO$_4$) using Pasteur pipettes of decreasing diameter. The neurons were pelleted by centrifugation (600×$g$ for 10 min at 4°C) and plated onto 8-mm glass coverslips pre-treated with Matrigel (BD Biosciences; San Jose, CA) in 48-well plate. Neurons were allowed to adhere for 30 min prior to adding 0.3 ml of pre-warmed culture medium consisting of Neurobasal A (Life Technologies; Carlsbad, CA), 2 mM GlutaMAX-I (Life Technologies, Carlsbad, CA), 2% B-27 supplement, and 5% FBS. After 4–12 hr, culture media was completely replaced with the same media without FBS. Neurons were incubated at 37°C/5% CO$_2$, and half of the medium was replaced with fresh medium on each of the first 3 days of culture. Neurons were kept in culture for 10–14 days prior to experiments.

## Somatodendritic GIRK current recordings

Coverslips containing neurons were transferred to a chamber containing a low-K$^+$ bath solution (in mM): 145 NaCl, 4 KCl, 1.8 CaCl$_2$, 1 MgCl$_2$, 5.5 D-glucose, 5 HEPES/NaOH (pH 7.4). Borosilicate patch pipettes (3–5 MΩ) were filled with (in mM): 130 KCl, 10 NaCl, 1 EGTA/KOH (pH 7.2), 0.5 MgCl$_2$, 10 HEPES/KOH (pH 7.2), 2 Na$_2$ATP, 5 phosphocreatine, 0.3 GTP. Baclofen (*R*-(+)-b-(aminomethyl)-4-chlorobenzenepropanoic acid hydrochloride) was purchased from Sigma (St. Louis, MO). Baclofen-induced currents were measured at room temperature using a high-K$^+$ bath solution (in mM): 120 NaCl, 25 KCl, 1.8 CaCl$_2$, 1 MgCl$_2$, 5.5 D-glucose, 5 HEPES/NaOH (pH 7.4). The high-K$^+$ bath solution (+/− baclofen) was applied directly to the soma and proximal dendrites with an SF-77B rapid perfusion system (Warner Instruments, Inc.; Hamden, CT).

Current responses to the application of the high-K$^+$ solution (+/− baclofen) were measured at a holding potential of −80 mV. Membrane potentials and whole-cell currents were measured in large neurons (>75 pF) with hardware (Axopatch-700B amplifier, Digidata 1440A) and software (pCLAMP v. 10.3) from Molecular Devices (Sunnyvale, CA). All currents were low-pass filtered at 2 kHz, sampled at 5 kHz, and stored on computer hard disk for subsequent analysis. Peak and steady-state current amplitudes were measured for each experiment. Current activation rates were extracted from a standard exponential fit of the current trace corresponding to the onset of drug effect and the peak evoked current, while deactivation rates were extracted from an exponential fit of the trace corresponding to the return of current to baseline following removal of drug (Clampfit v. 10.3 software). Current desensitization was defined as % change in steady state current from the maximal baclofen-evoked response amplitude during 10 s of continuous drug application. Only experiments where access resistances were stable and low (<20 MΩ) were included in the analysis.

## Immunogold electron microscopy

Immunohistochemical reactions were carried out using the pre-embedding immunogold method as described earlier (*Lujan et al., 1996*). Briefly, after blocking with 10% serum for 1 hr at room temperature free-floating sections were incubated for 48 hr with anti-RGS7 antibodies (1–2 mg/ml). Sections were washed and incubated for 3 hr with goat anti-rabbit IgG coupled to 1.4 nm gold (Nanoprobes

Inc) at 1:100 dilution. Sections were washed, postfixed in 1% glutaraldehyde and processed for silver enhancement of the gold particles with an HQ Silver kit (Nanoprobes Inc.). The reacted sections were treated with osmium tetraoxide (1% in 0.1 M PB), block-stained with uranyl acetate, dehydrated in graded series of ethanol and flat-embedded on glass slides in Durcupan (Fluka) resin. Regions of interest were cut at 70–90 nm on an ultramicrotome (Reichert Ultracut E; Leica). Staining was performed on drops of 1% aqueous uranyl acetate followed by Reynolds's lead citrate. Ultrastructural analyses were performed in a Jeol-1010 electron microscope.

To establish the relative the abundance of RGS7 immunoreactivity along the plasma membrane of pyramidal cells, we used 60-μm coronal slices processed for pre-embedding immunogold immunohistochemistry. The procedure was similar to that used previously (*Lujan et al., 1996*). Briefly, for each of three animals from different postnatal ages and adult, three samples of tissue were obtained for preparation of embedding blocks (totalling nine blocks for each age). To minimize false negatives, electron microscopic serial ultrathin sections were cut close to the surface of each block, as immunoreactivity decreased with depth. We estimated the quality of immunolabelling by always selecting areas with optimal gold labelling at approximately the same distance from the cutting surface. Randomly selected areas were then photographed from the selected ultrathin sections and printed with a final magnification of 45 000X. Quantification of immunogold labelling was carried out in reference areas totalling approx. 1,800 μm$^2$ for each age. Immunoparticles identified in each reference area and present in different subcellular compartments (dendritic spines, dendritic shafts and somata) were counted. We measured the radial distance of each immunoparticle to the plasma membrane, being 0 for those just located in the plasma membrane. The data was expressed as percentage of immunoparticles along the radial distance from the plasma membrane expressed in nanometers.

## Hippocampal slices

Mice were sacrificed under isoflurane anesthesia, and brains were rapidly removed and placed in ice-cold artificial cerebrospinal fluid (aCSF) without $CaCl_2$, composed of (mM): 124 NaCl, 3 KCl, 24 $NaHCO_3$, 1.25 $NaH_2PO_4$, 1 $MgSO_4$, and 10 D-Glucose, equilibrated with 95% $O_2$ and 5% $CO_2$. The tissue was cut in 350–400 μm thick sections with a Vibrating microtome (Leica VT1200S, Germany). The slices were warmed to 35°C for 25–45 min in aCSF supplemented with 2 mM $CaCl_2$, and equilibrated with 95% $O_2$ and 5% $CO_2$. Then slices were maintained in gassed aCSF at room temperature until being transferred to submerged-type recording chambers of volume ~1.5 ml. Here, the slices were constantly superfused (1–2 ml/min) with warmed (30–31°C), gassed aCSF. All measurements were performed by an experimenter blind to genotype.

## Patch clamp recordings in slices

CA1 neurons were visually identified in the hippocampal transverse slices of 350 μM thickness using Scientifica SliceScope system. Glass microelectrodes with an open-tip resistance of 3.5–6.5 MΩ were used. The following internal solution was used (mM): 130 K-Gluconate, 20 KCl, 10 K-HEPES, 0.2 EGTA, 0.3 Na-GTP and 4 Mg-ATP (pH 7.3). To determine intrinsic cellular properties such as resting membrane potential, input resistance and spike numbers, 500 ms, 50 pA, 10-step hyperpolarizing and depolarizing current injections were delivered every 10 s. Cells with series resistance >20 MΩ or resting membrane potentials > −55 mV were excluded from analysis. Liquid junction potential was −14 mV. Spontaneous EPSCs (sEPSCs) were measured by holding the cells at −70 mV in normal aCSF. At least 100 events, which are above 4 pA, were obtained during sEPSC measurements in each cell.

## Synaptic plasticity measurements

Field excitatory postsynaptic potentials (fEPSPs) were elicited by a concentric bipolar stimulating electrode (inner diameter [ID]: 25 μm; outer diameter [OD]: 125 μm, FHC Inc., Bowdoin, ME) connected to a constant current isolated stimulator unit (A-M Systems; Carlsborg, WA) and recorded with low resistance (3–5 MΩ) glass pipettes (ID: 1.16 mm, OD: 1.5 mm, Harvard Apparatus, Holliston, MA) filled with aCSF. The electrodes were placed in the *stratum radiatum* of the CA1 area of the dorsal hippocampus slices. Stimulation frequency was set to 0.05 Hz. Input-output curves were generated by adjusting the stimulus intensity in increments of 10 μA, from 0 to 100 μA. Paired pulse ratio (PPR) was assessed using a succession of paired pulses separated by time intervals ranging from 25 to 1000 ms, delivered every 20 s. The degree of facilitation was determined by taking the ratio of the initial slope of the second fEPSP to the initial slope of the first fEPSP. A PPR >1 was considered to reflect facilitation. For synaptic plasticity experiments, a stable baseline for at least 30 min was achieved prior to

induction. LTP and LTD were recorded for 1 hr after HFS (2 tetanized stimuli (TS) of 100 Hz for 1s each) or LFS (2 Hz for 10 min, 1200 pulses) was applied. Depotentiation was achieved by applying HFS followed by LFS after 1-2 min interval.

## Behavioral analysis

Spatial learning and memory were evaluated in the Morris water maze using a video tracking systems (EthoVision XT, Noldus Information Technology, Wageningen, Netherlands). Mice first received 4 cued trials (visible but variable platform location) on the first day to determine if non-associative impairment in other behavioral responses affecting performance in this task, such as exploratory activity, motor coordination, vision and motivation. After completing the cued trials, spatial learning acquisition was evaluated during the place condition (hidden platform, constant location; 4 trials/day, 6 consecutive days). Escape latency and success rates in finding hidden platform were calculated for all place trials. Retention performance was evaluated during probe trials (platform removed), which were conducted 24 hr after the last cued trial on the sixth day. Latency to the first target platform area crossing and the number of crossings in the probe trial (1 min) served as the dependent variables.

For fear conditioning experiments, mice were habituated in individual conditioning chambers to obtain freezing baseline on the first day. On the second day, mice were trained in individual conditioning chambers (Med Associates, St. Albans, VT). Video images were recorded via video tracking systems (EthoVision XT, Noldus Information Technology, Wageningen, Netherlands). In context A, visible light was turned on, a stainless steel grid floor inserted, and the chambers were cleaned with 70% ethanol prior to conditioning. In context B, white plastic inserts were placed inside the chamber to change the shape, size and texture of the wall and the floor. A small weight boat with orange extract is placed behind the wall inserts to provide a novel smell for the chambers. The chambers were cleaned with 70% isopropanol. During the training sessions, mice were placed in context A and allowed to explore for 3 min prior to the delivery shocks (0.75 mA, 1 s). We used auditory tone (85 dB, 30 s duration) as CS and an electric foot shock (delivered through the grid floor, 0.75 mA AC current, 1 s duration) as US. Mice received 3 tone-shock stimuli (with 1–2 min interval) during the training day. In every session, tone always co-terminated with electric shock. Mice were removed from the conditioning chamber 30 s after the last shock. Contextual test (5 min) was done in context A 24 hr after training. Cue test was performed 1 hr after contextual test in context B. The animals were brought into the test room in a different covered mover with different bedding inside to avoid contextual reminders. The animal was immediately placed in the contextual B chamber and allowed to explore for 3 min prior to the delivery the auditory tone (85 dB) for another 3 min. The freezing response was measured with the automated tracking and analyzed offline.

For the novel object recognition test, mice were habituated to the arena for two consecutive days, where they were placed in a white plastic open-box of dimensions 43.2 cm2 by 30.5 cm height for 10 min each. The acquisition phase of the object recognition assay involved placing each individual mouse in the test arena for 8 min in the presence of two identical objects (either silver or black objects in different shape, size and texture, the order of which alternated between mice). The video was recorded via a camera. The time spent investigating both identical objects was recorded by automatic tracking system (EthoVision XT, Noldus Information Technology, Wageningen, Netherlands). 24 hr after the acquisition phase, each mouse was reintroduced to the test arena for 8 min. During this phase (retention testing phase) the mouse was presented with one copy of the object that it was exposed to during the acquisition phase (familiar object) and one novel object (either silver or black objects, depending on which was presented during the acquisition phase). The time spent investigating each object was recorded. All objects and the testing box were thoroughly cleaned with 70% methanol between mice to remove any odor cues.

## Measurements of G$\beta\gamma$ interactions with effector by BRET assay

Agonist-dependent cellular measurements of bioluminescence resonance energy transfer (BRET) between Venus-Gβ1γ2 and its effector fragment masGRK3ct-Nluc were performed upon reconstitution in living cells as previously described with slight modification (*Hollins et al., 2009*; *Masuho et al., 2013*). The masGRK3ct-Nluc construct contained amino acids G495-L688 of bovine GRK3 (NP_776925), preceded by a myristic acid attachment peptide (mas; MGSSKSKTSNS). The stop codon of GRK3 was replaced with a GGGS linker, which was followed by the NanoLuc (Nluc) (*Hall et al., 2012*). Briefly, GABA$_{B1}$R, GABA$_{B2}$R, GαoA, Venus156-239-Gβ1, and Venus1-155-Gγ2 constructs were transfected

into HEK293T/17 cells at a 1:1:2:1:1 ratio with increasing masGRK3ct-Nluc from 0.125 to 6 in ratio. 5 μg total DNA was delivered per $4 \times 10^6$ cells in a 6-cm-dish. 16–24 hr post transfection cells were stimulated with 100 μM GABA followed by treatment with 100 μM CGP 54626. The BRET signal is determined by calculating the ration of the light emitted by the Venus-Gβ1γ2 (535 nm) over the light emitted by the masGRKct-Rluc8 (475 nm). The average baseline value recorded prior to agonist stimulation was subtracted from BRET signal values, and the resulting difference (ΔBRET) was obtained.

### Data analysis

Statistical analyses were performed using Prism (GraphPad Software, Inc.; La Jolla, CA). Data are presented throughout as the mean ± SEM. Student $t$ test, one-way or two-way ANOVA, followed by Bonferroni's *post hoc* test were used as appropriate. The minimal level of significance was set at $p<0.05$.

## Acknowledgements

We thank Dr Orlandi for his help with preparing samples for the EM studies, Drs Clement (Chelliah), Kourrich and Ozkan for technical advice and discussions and Mrs Martemyanova for help with animal breeding and genotyping. This work was supported by NIH grants DA021743 (KAM), DA026405 (KAM), MH061933 (KW), DA034696 (KW), HL105550 (KW and KAM) and by grants from the Spanish Ministry of Education and Science (BFU-2012-38348) and CONSOLIDER (CSD2008-00005) to RL.

## Additional information

### Funding

| Funder | Grant reference number | Author |
| --- | --- | --- |
| National Institutes of Health | DA021743 | Kirill A. Martemyanov |
| National Institutes of Health | DA026405 | Kirill A. Martemyanov |
| National Institutes of Health | MH061933 | Kevin Wickman |
| National Institutes of Health | DA034696 | Kevin Wickman |
| Spanish Ministry of Education and Science | BFU-2012-38348 | Rafael Lujan |
| CONSOLIDER | CSD2008-00005 | Rafael Lujan |
| National Institutes of Health | HL105550 | Kirill A. Martemyanov |

The funders had no role in study design, data collection and interpretation, or the decision to submit the work for publication.

### Author contributions

OO, KX, Acquisition of data, Analysis and interpretation of data, Drafting or revising the article; IM, AF-S, RL, Acquisition of data, Analysis and interpretation of data; KW, KAM, Conception and design, Analysis and interpretation of data, Drafting or revising the article

### Ethics

Animal experimentation: This study was performed in strict accordance with the recommendations in the Guide for the Care and Use of Laboratory Animals of the National Institutes of Health. All of the animals were handled according to approved institutional animal care and use committee (IACUC) protocols (#08-133) of the Scripps Research Institute.

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
