## [Decision Letter]

Thank you for sending your work entitled “Rgs7/Gβ5/R7BP complex regulates synaptic plasticity and memory by modulating hippocampal GABA_B_R-GIRK signaling” for consideration at *eLife*. Your article has been favorably evaluated by a Senior editor and 3 reviewers, one of whom is a member of our Board of Reviewing Editors.

The following individuals responsible for the peer review of your submission have agreed to reveal their identity: Richard W Aldrich (Reviewing editor); David Clapham (peer reviewer).

The Reviewing editor and the other reviewers discussed their comments before reaching this decision, and the Reviewing editor has assembled the following comments to help you prepare a revised submission.

Ostrovskaya et al. continue their investigation of the hippocampal metabotropic GABAB activation of the GIRK (Kir3) channel. This pathway is initiated by gamma-aminobutyric acid neurotransmitter binding to its G protein coupled receptor, GABAB. This initiates Gβγ activation of Kir3 heterotetramers, which hyperpolarize neurons. RGS proteins (30 members) regulate GPCR coupling to effectors.

In this paper, the authors use knockout mice to suggest that both *Rgs7*^*-/-*^ and *R7bp*^*-/-*^ are in a complex with the GABA-GIRK axis to dampen the response to GABA agonist in hippocampus. Deletion of *Rgs7* in mice sensitizes GIRK responses to GABAB receptor agonist and slows channel deactivation kinetics. CA1 pyramidal neurons lacking *Rgs7* are thus more hyperpolarized and less excitable. Interestingly, GIRK onset kinetics (but not deactivation rates) were affected by *Rgs7* ablation only at low agonist concentrations while *R7bp* ablation significantly affected deactivation kinetics only at higher agonist concentrations. Basal excitatory transmission and excitatory presynaptic function (inferred from FV and PPR) are not changed in *Rgs7*^*-/-*^ mice. Loss of R7bp results in ineffective targeting, and consequent decrease Rgs7 function. The authors propose that Rgs7/Gβ5/R7bp adjusts signaling strength in the hippocampal GABAB-GIRK pathway. This disrupts LTD and depotentiation, and this somehow interferes with associative learning and formation of contextual memory.

The following issues must be adequately addressed for a revised version to be accepted:

1) Some may consider that the results overlap with the authors' paper in Nature Neuroscience in 2010 – “β5 recruits R7 RGS proteins to GIRK channels to regulate the timing of neuronal inhibitory signaling”. We realize that there are new elements, but it might be good to state more clearly what you showed before and what you show now.

2) Readers will be confused by Figure 7. You need to explain that the GGL domain of RGS7 binds Gβ5 in brain. Also, what do the excess subunits at high [agonist] bind to in order to change deactivation kinetics and response sensitivity?

3) It is not valid to conclude much about activation rates from measurements of “rising” phases when the records also decay. In this case the rising phase contains contributions from both activation and inactivation. If a treatment changes the decaying phase also, then conclusions about activation are invalid, as the two processes cannot be well separated. The treatment of activation rates needs to take this into account, at least by some textual qualification.

4) Readers would appreciate some more explanation of how increasing LTD and depotentiation are needed for memory. If no one knows how, say so.

5) Has the lag been previously reported in the Gβ5 KO or R7BP KO? If not, what explains the difference and the presence of the lag in the current work? Do the authors have independent evidence for enhanced levels of free Gβγ relative to GIRK? Also, it seems that there is a 'hook' before the lag in the RGS7 KO. Is this reproducible? How can this be explained by changes in Gβγ activation?

6) RGS7 is widely expressed and knocked-out in the whole mouse. Some of the changes in behavior could be due to changes in signaling outside of the hippocampus or in multiple brain regions. This needs to be discussed.

7) It is not readily apparent how the ec50 and slope change in R7BP KO, but only the EC_50_ changes in RGS7 KO (which also has decreased R7BP). Why would the cooperativity increase in the absence of R7BP? Modeling the currents with different scenarios for G protein turnover could be very helpful. At minimum this should be thoroughly discussed.

---

## [Author Response]

*1) Some may consider that the results overlap with the authors' paper in Nature Neuroscience in 2010* – *“β5 recruits R7 RGS proteins to GIRK channels to regulate the timing of neuronal inhibitory signaling”. We realize that there are new elements, but it might be good to state more clearly what you showed before and what you show now*.

In our previous study, we established that RGS-Gβ5 complexes play an essential role in modulating GABA_B_R-GIRK signaling. To reach this conclusion, we examined the phenotype of mice lacking Gβ5, a subunit of the complexes that could involve four different RGS proteins in the R7 RGS subfamily: RGS6, RGS7, RGS9 or RGS11. We performed analysis of GIRK kinetics in primary hippocampal cultures, recorded evoked sIPSCs in hippocampal slices, and examined the pharmacological sensitivity of mice lacking Gβ5 to baclofen. The identity of the RGS protein within the Gβ5 complex that mediates regulation of hippocampal GABA_B_R-GIRK signaling remained unknown. Further unknown was whether another subunit of the complex, R7BP, was involved in GABA_B_R-GIRK signaling regulation. Finally, the relevance of observed alterations in GIRK channel function to the physiology of the intact hippocampal circuit and synaptic plasticity has not been explored. The current study fills these gaps by identifying RGS7 to be the main RGS isoform in the Gβ5-RGS complexes that regulate GABA_B_R-GIRK signaling in the hippocampus, by establishing the role of R7BP subunit in this regulation, and by demonstrating that these complexes play a key role in synaptic plasticity and memory. We have further clarified the distinction between our prior and this current study in the Introduction and Discussion sections of the revised manuscript.

*2) Readers will be confused by*
Figure 7*. You need to explain that the GGL domain of RGS7 binds Gβ5 in brain. Also, what do the excess subunits*
*at high [agonist] bind to in order to change deactivation kinetics and response sensitivity?*

In order to avoid unnecessary confusion, we decided to eliminate Figure 7 in the revised manuscript. Cartoon representations are inherently limited by the restrictive nature of the graphic space. As a result, we found that attempts to emphasize one aspect of the study resulted in an unwarranted misperception to other issues. However, we did leave the verbal description of the model in the Discussion section of the manuscript.

*3) It is not valid to conclude much about activation rates from measurements of “rising” phases when the records also decay. In this case the rising phase contains contributions from both activation and inactivation. If a treatment changes the decaying phase also, then conclusions about activation are invalid, as the two processes cannot be well separated. The treatment of activation rates needs to take this into account, at least by some textual qualification*.

We acknowledge that the analysis of the “activation” phases of the GIRK channel responses could be complicated. In fact, slowing the activation phase has repeatedly been shown to be related to deceleration of GTP hydrolysis rates in the absence of RGS proteins. However, from the channel perspective, upon agonist application, the steady-state of the G protein cycle is shifted tremendously towards favoring channel opening with no appreciable channel inactivation, which becomes prominent only when the agonist is removed. Indeed, our additional analysis reveals no significant genotype differences in current inactivation during baclofen application (desensitization). In our mind, this allows a rather uncomplicated analysis of the rising phases that do not seem to be contaminated much by channel inactivation. We agree with the reviewers’ point, however, that analysis of the activation kinetics may not allow us to conclude much and we further qualified this in the revised manuscript. We did not put much of an emphasis on analysis of activation kinetics. Instead, our model mostly relies on sensitivity (steady-state currents) and the response deactivation, which directly correlates with the GTPase activity of the G proteins, and as a result is much more interpretable.

*4) Readers would appreciate some more explanation of how increasing LTD and depotentiation are needed for memory. If no one knows how, say so*.

We supplied more information regarding the relevance of LTD and depotentiation to learning and memory.

*5) Has the lag been previously reported in the Gβ5 KO or R7BP KO? If not, what explains the difference and the presence of the lag in the current work? Do the authors have independent evidence for enhanced levels of free Gβγ relative to GIRK? Also, it seems that there is a 'hook' before the lag in the RGS7 KO. Is this reproducible? How can this be explained by*
*changes in Gβγ activation?*

The lag has been described before for the wild-type neurons and noted to be present in both activation and deactivation phases (Otis 1993, JPhysiol; Sodickson 2006, J Neurosci; Zhang 2002, J Physiol). To the best of our knowledge, the present study for the first time reports an analysis of changes in the lag time induced by the loss of RGS protein complex components. To obtain independent evidence that this lag is related to changes in Gβγ stoichiometry relative to the effector (e.g., the GIRK channel), we modeled changes in Gβγ to effector ratios in a reconstituted system using a BRET-based reporter assay. The results of these experiments are entirely consistent with electrophysiological recordings of GIRK channel kinetics in the native system and show increase in response deactivation lag time upon an increase in Gβγ stoichiometry over an effector. These results are now reported in the revised version of the manuscript (Figure 3). The “hook” mentioned by the reviewer is a recording artifact that sometimes affects responses upon fast solution change in our perfusion setup. It randomly affects cells regardless of their genotype. In the revised manuscript, we substituted the traces that showed a prominent “hook” with cleaner traces.

*6) RGS7 is widely expressed and knocked-out in the whole mouse. Some of the changes in behavior could be due to changes in signaling outside of the hippocampus or in multiple brain regions. This needs to be discussed*.

This is an important point. At present, we cannot exclude the possibility that some behavioral changes observed in this study may be contributed by changes in other brain regions. We have added the discussion of this point to the revised manuscript.

*7) It is not readily apparent how the ec50 and slope change in R7BP KO, but only the EC*_*50*_
*changes in RGS7 KO (which also has decreased R7BP). Why would the cooperativity increase in the absence of R7BP? Modeling the currents with different scenarios for G protein turnover could be very helpful. At minimum this should be thoroughly discussed*.

First, we would like to point out that elimination of either R7BP or RGS7 changes the EC_50_ for GABA_B_R-GIRK signaling. As already discussed in the paper, and entirely consistent with biochemical characterization, the total loss of RGS7 (that also leads to reduction of associated R7BP) results in the biggest EC_50_ change – from 1.48-2.14 μM for wild-type to 0.35- 0.58 μM for *RGS7*^*-/-*^. However, knockout of R7BP, which affects RGS7 in subtler way (via membrane localization and activity regulation) also results in a significant EC_50_ reduction (to 0.65-0.81 μM for *R7BP*_*-/-*_; 95 % CI). We feel that the extent of the collected data and insufficient quantitative knowledge of the key parameters for protein/protein interactions in the GABA_B_R- GIRK signaling cascade significantly limits our ability to build an accurate model with a significant predictive value without doing substantially more work. On an intuitive level, however, we think that the observed increase in the cooperativity of GIRK channel activation upon R7BP loss may reflect larger proportion of Gβγ subunits that reach an effector (GIRK) while being deactivated in its vicinity by RGS7/Gβ5 bound to the channel. By physically binding to GIRK channel RGS7 alone, but not its complex with R7BP, may further act to promote productive interactions of Gβγ (and/or Gα) with the GIRK channel, thus increasing cooperativity of its activation. We have discussed these possibilities in the revised manuscript.